# Dietary Patterns and Associations with Myopia in Chinese Children

**DOI:** 10.3390/nu15081946

**Published:** 2023-04-18

**Authors:** Chunjie Yin, Qian Gan, Peipei Xu, Titi Yang, Juan Xu, Wei Cao, Hongliang Wang, Hui Pan, Zhibin Ren, Hui Xiao, Kai Wang, Ying Xu, Qian Zhang

**Affiliations:** 1School of Public Health, Xinjiang Medical University, Urumqi 830017, China; yinchunjie2020@163.com (C.Y.);; 2Chinese Center for Disease Control and Prevention, NHC Key Laboratory of Trace Element Nutrition, National Institute for Nutrition and Health, Beijing 100050, China; 3School of Medical Engineering and Technology, Xinjiang Medical University, Urumqi 830017, China

**Keywords:** dietary pattern, myopia, child, factor analysis, FFQ, cross-sectional study

## Abstract

Dietary shifts in populations undergoing rapid economic transitions have been proposed as partly contributing toward the rapid intergenerational rise in myopia prevalence; however, empirical evidence of the effect of dietary factors on myopia is limited. This study investigated the association between dietary factors and incident myopia in Chinese children aged 10–11 years. We evaluated dietary habits using a 72-item food frequency questionnaire (FFQ) among 7423 children. Myopic status was assessed using the “General Personal Information Questionnaire”. Principal component analysis was used to extract dietary patterns and investigate their association with myopia. After adjusting for potential confounders, participants with the highest adherence to dietary pattern A (95% confidence interval (CI): 0.66–0.92, *p* for trend = 0.007) and dietary pattern C (95% CI: 0.58–0.80, *p* for trend < 0.001) had a lower risk of myopia than participants with the least adherence. Both of these dietary patterns are characterized by high consumption of meats, aquatic product, dairy and its products, eggs, legumes, vegetables, fruits, grains, and potatoes. Our findings suggest that other environmental factors, such as those related to the dietary environment, may contribute to the development of myopia. These findings can serve as a reference for diet-related primary prevention of myopia.

## 1. Introduction

Myopia, or nearsightedness, is the most common refractive error (RE) in children and young adults [1]. A refractive error in which rays of light entering the eye parallel to the optic axis are brought to a focus in front of the retina when ocular accommodation is relaxed [2]. This is usually caused by the eyeball being too long from front to back, an overly curved cornea, and/or a lens with increased optical power [2]. Compared with the cornea and crystalline lens, an increased axial length (AL) is an important factor related to refractive error (RE) progression [3]. When considering quantitative thresholds and descriptions of myopia, the following quantitative definitions are proposed. Myopia is ‘‘a condition in which the spherical equivalent objective refractive error is ≤−0.50 D’’. Low myopia is “a condition in which the spherical equivalent objective refractive error is ≤−0.5 and >−6.00 D”. High myopia is “a condition in which the spherical equivalent objective refractive error is ≤−6.00 D” [2]. High myopia carries the highest risk of glaucoma (open-angle); cataract (nuclear, cortical and posterior subcapsular); retinal tears, which may lead to retinal detachment (RD); and myopic maculopathy or myopic macular degeneration (MMD) [4]. Although low and moderate myopes are less likely to develop such a severe visual outcome, they are at significant risk of developing MMD, RD, cataract, glaucoma (open-angle), and migraine [5,6]. In East and Southeast Asia, the prevalence of myopia at the age of 17–18 has reached 80–90% [7]. The prevalence of myopia is expected to grow, which is predicted to affect approximately half of the world’s “population by 2050 [8]”. As the incidence of myopia increases significantly, the economic burden related to the increase in myopia rises [9]. Therefore, identifying modifiable etiology and instituting preventive measures early in life is now more important. Numerous studies have described the involvement of both genetic and environmental components in causing myopia [1,10]. Genetic factors may contribute to myopic differences [11]. However, it is important to recognize the potential role of shared environments. The list of environmental factors influences on myopia development studied is long, including education, near work, outdoor activity, visual environment, retinal defocus patterns, eye growth, etc. [12]. Nonetheless, our understanding of the environmental factors influencing myopia progression remains elusive. Young et al. [13] examined 41 family units (197 participants) and found the correlation among siblings was highest. Moreover, despite virtually no myopia in parents or grandparents, the prevalence of myopia in children was approximately 50%. They speculated that the advent of formal (classroom) education and/or recent dietary changes for the population might be to blame.

Dietary shifts in populations undergoing rapid economic transitions have been proposed to partly contribute to the rapid intergenerational increase in the prevalence of myopia [14,15]. Puberty is a dynamic period of growth characterized by rapid alterations in body size, shape, and composition [16]. Veysi et al. [17] found that eye growth may be more vulnerable to environmental factors during puberty. Several studies have reported that food and nutrient intake is closely related to the development of myopia. Evidence from children aged 11–14 years in China suggests that the consumption of sugary foods, such as cakes, preserved fruits, candies, chocolate, and ice cream, is positively correlated with the prevalence of myopia [18]. In a cross-sectional epidemiological study of 586 children aged 6–12 years by Zhuzhu et al., wholegrain intake (>50%) was an independent protective factor against myopia, especially in children aged 9–12 years [19]. In a cohort study of young children, Willem et al. [20] found a significant association between serum 25(OH)D levels, AL, and myopia. Serum 25(OH)D is derived from Cholecalciferol (vitamin D3) and Ergocalciferol (vitamin D2). Vitamin D3 is formed in the skin after sunlight exposure, and absorbed by the gut after dietary intake of, e.g., fatty fish. Vitamin D2 results from intake of foods containing yeasts and fungi. However, these links remain poorly understood due to the paucity of population-based dietary assessment data.

We aimed to assess the association between dietary factors, using a validated food frequency questionnaire (FFQ) and incident myopia in Chinese children aged 10–11 years based on data from the Survey and Application of the Nutrition and Health System for Children Aged 0–18 Years in China (2017–2021) [21].

## 2. Methods

### 2.1. Participants

The data for this study were obtained from the National Nutrition and Health Systematic Survey and Application for 0–18-year-old children. Children 10–11 years of age from 14 provinces (Zhejiang, Jiangxi, Shanxi, Beijing, Hunan, Henan, Guangdong, Guangxi, Yunnan, Chongqing, Shanxi, Qinghai, Jilin, and Liaoning) and 28 counties in China were surveyed using a multistage stratified random sampling method. Each year, there were 196 students, half of which were boys and half of which were girls. A total of 10976 children aged 10–11 were surveyed. This study was approved by the Ethics Committee of the Institute of Nutrition and Health of the Chinese Center for Disease Control and Prevention (Grant No. 2019-011), and all parents of the surveyed children signed an informed consent form.

### 2.2. Basic Information Interview

Data were collected using the “General Personal Information Questionnaire”. The questionnaire was filled out by the children themselves after detailed explanation by trained investigators. The main content of each questionnaire was as follows.

The “General Personal Information Questionnaire” included basic information, such as myopia status, type of survey site, date of birth, sex, etc. The myopia status information was collected from a series of questions, such as “Do you currently have myopia?”; “What have you done to correct your vision?”; and “At present, if you wear glasses, what is the diopter?”

### 2.3. Dietary Assessment

Dietary information was assessed using a semi-quantitative FFQ, which collected average habitual dietary intake during the previous month. A total of 72 food items in the FFQ were categorized into 11 food groups based on similarities in nutrient composition or culture as follows: (1) grains and potatoes; (2) legumes; (3) vegetables; (4) mycorrhizae; (5) fruits; (6) dairy and its products; (7) meats; (8) aquatic products; (9) eggs; (10) snacks; and (11) beverages. Reporting options for the child’s usual portion sizes in grams and consumption frequency in the previous month included “never” (recorded as zero) or the number of times, i.e., “per day”, “per week”, or “per month”.

#### Dietary Pattern

The 11 groups in the FFQ were used to explore dietary patterns using principal component analysis (PCA) with orthogonal (varimax) rotation. The Kaiser–Meyer–Olkin and Bartlett’s tests were used to determine whether the data were suitable for PCA. Three dietary patterns were identified according to factors with eigenvalues greater than 1, combined with the scree plot breakpoints and cumulative variance contributions. Regression was used to calculate the factor scores, and the higher the score, the more the subject tended to follow this dietary pattern; according to the quartiles of the factor scores, the subjects were divided into Q1, Q2, Q3, and Q4 groups in descending order. Q1 indicates the least tendency for this dietary pattern, and Q4 indicates the highest tendency for this dietary pattern.

### 2.4. Physical Activity Questionnaire

The “physical activity questionnaire” included daily screen time, the average daily sleep duration for a week, and days of moderate-to-high intensity physical activity per week. Information on daily screen time included the total amount of time spent watching TV and using cell phones, desktop computers, and tablets. Average daily sleep duration per week was calculated using daily sleep time on weekdays and daily sleep time on rest days. Moderate-to high intensity physical activity per week was determined using the number of days when more than 1 h per day was spent on physical activity. The reliability and validity of the “physical activity questionnaire” were tested [22,23].

The “General Personal Information Questionnaire”, FFQ, and “physical activity questionnaire” were designed by the project team and revised after expert demonstration and preliminary investigation. The questionnaire was explained by well-trained investigators and then completed by the students.

### 2.5. Physical Examination

The height and weight of the children were measured uniformly by trained and qualified investigators in each survey county/district.

All children were weighed in the morning after fasting using a digital weight scale (GMCS-I electronic scale; Jianmin, Beijing, China). The maximum capacity of the scale was 100 kg, and the smallest division was 100 g. 

The heights of all children aged 10–11 years were determined using a stadiometer (accuracy, 0.1 cm; Jianmin).

BMI was calculated based on the formula: body weight/the square of height. Participants with BMI > 35 or <10 were excluded from the analysis.

### 2.6. Covariates

The variables used for multiple adjustments in the logistic regression analysis were defined as follows. (1) Living residence was divided into urban and rural areas. (2) The living regions were separated into southern and northern China. According to academic consensus, China is divided into northern and southern China by the Qinling Mountains to the west and by the Huai River (one of the Yangtze tributaries) to the east [24]. (3) Nutrition status was categorized as stunting and wasting, normal weight, overweight, and obese. Stunting and wasting classification was based on the Screening Standard For Malnutrition of School-aged Children and Adolescents (WS/T 456-2014) [25] in China. The overweight and obesity classification was based on the screening for overweight and obesity among school-aged children and adolescents (WS/T 586-2018) [26] in China. (4) Medium- and high-intensity physical activity was categorized based on weekly total days as 0 d/week, 1 d/week, 2 d/week, and ≥3 d/week. (5) Screen time was categorized based on total minutes per day as 0 min/day, 0–59 min/day, and ≥60 min/day. (6) Sleep duration was categorized as ≤9 h/day or over 9 h/day.

### 2.7. Statistical Analysis

All statistical analyses were conducted using SAS version 9.4 software (SAS Institute, Inc., Cary, NC, USA) and SPSS v23 (IBM, Chicago, IL, USA). Categorical variables are presented as frequencies and percentages, and the statistical difference was determined using the chi-squared test. Binary logistic regression was used to examine the association between each dietary pattern and myopia. Myopia was used as a dependent variable, and the scores of each dietary pattern in quartiles were used as independent variables. The results were given by odds ratios (ORs) and 95% confidence intervals (CIs) compared with the reference quartile (the lowest one) in different models. Model 1 was adjusted for residence and region. Model 2 was adjusted for obesity status, medium- and high-intensity physical activity, screen time, and sleep duration. In addition, all *p* values for linear trends were calculated based on the quartiles of the dietary patterns.

## 3. Results

### 3.1. Characteristics of Participants

The baseline characteristics of the 7423 children aged 10–11 years with dietary patterns, physical activity, physical examination, and myopia data are shown in Table 1. The sample included 1746 (23.5%) children with myopia and 5677 (76.5%) without. Subjects with myopia were more likely to be older (27.3% and 19.7%; *p* < 0.001, respectively), girls (girl: 26.3% vs. 20.8%; *p* < 0.001), living in urban areas (urban: 31.1% vs. 18.0%; *p* < 0.001) and northern China (northern China: 25.7% vs. 21.2%; *p* < 0.001), and overweight or obese (overweight and obesity: 26.0% vs. 22.6% vs. 18.3%; *p* < 0.001). However, there were no significant differences in medium- and high-intensity physical activity, screen time, or sleeping duration between children with myopic and without myopia.

### 3.2. Dietary Patterns among Chinese Children

Three dietary patterns met the retention criteria of eigenvalues greater than 1 and explained 46.23% of the variance in dietary consumption. The food groups and factor loadings for each dietary pattern are presented in Table 2. Dietary pattern A, which explained 16.55% of the variance, was identified by high factor loadings for (and more frequent consumption of) meats, aquatic product, dairy and its products, eggs, and legumes. Dietary pattern B, which explained 15.59% of the variance, was characterized by high factor loadings for snacks, beverages, and mycorrhizae. Dietary pattern C, which accounted for 14.20% of the variance, was defined by high factor loadings for vegetables, fruit, grains, and potatoes. 

The general characteristics of the study participants according to the three dietary pattern scores are shown in Appendix A. Subjects with high scores (Q4) for dietary pattern A more commonly lived in urban and southern China, were overweight or obese, and had a screen time of 0–60 min/d in comparison to those with low scores (Q1). In addition, subjects with high scores for dietary pattern B were more likely to be older, living in rural and northern China, with medium- and high-intensity physical activity of 0 d/w and 2 d/w, and screen time of ≥60 min/d than the subjects with low scores. Subjects with high scores for dietary pattern C more commonly lived in northern China, were overweight or obese, engaged in medium-and high-intensity physical activity with 0 d/w, and sleep duration > 9 h/d compared to those with low scores for dietary pattern C. There were no significant differences in sex across quartiles for any dietary pattern. 

### 3.3. Association between Dietary Patterns and Myopia

To evaluate the association between myopia and the three dietary patterns, we analyzed alterations in the three dietary patterns across the quartiles of the dietary pattern scores (Table 3). For dietary pattern A, those in the highest three quartiles of adherence had a significantly decreased risk of myopia than those with the lowest intake of foods associated with this dietary pattern; for those with the highest adherence quartile compared to the lowest adherent, the risk of myopia decreased by 22.5% (Q4 vs. Q1, OR = 0.69, 95% CI: 0.59–0.81, *p* for trend < 0.001). These differences remained significant in adjusted Models 1 and 2, where the most adherent quartile in both had decreased by 22% (95% CI 0.66, 0.91; *p* = 0.002 and 95% CI 0.66, 0.92; *p* = 0.003) compared to the least adherent reference category. The association between adherence to dietary pattern A and myopia was significant in both unadjusted models and after adjusting for sex, age, residency, region, nutritional status, medium-and high-intensity physical activity, screen time, and sleep duration (*p* for trend < 0.05).

For dietary pattern B, those in the highest two quartiles of adherence had no significantly decreased risk of myopia compared with those in the reference category. These results remained insignificant for the unadjusted model (*p* > 0.05), adjusted Model 1 (*p* > 0.05), and adjusted Model 2 (*p* > 0.05), compared to the least adherent reference category. Therefore, the association between higher food intake, dietary pattern B, and myopia was not significant.

For dietary pattern C, characterized by high consumption of vegetables, fruits, grains, and potatoes, those with the highest three quartiles of adherence had a significantly reduced risk of myopia than those with the lowest quartiles intake of foods associated with this dietary pattern (the most adherent quintile: OR = 0.71, 95% CI: 0.60–0.82, *p* for trend < 0.001) compared to the least adherent quintile (reference category). The association between adherence to dietary pattern C and myopia was significant for both the unadjusted and adjusted Models 1 and 2 (all *p* for trend < 0.001).

## 4. Discussion

In the present study, we extracted three a posteriori dietary patterns from the FFQs of our subjects, namely dietary patterns A, B, and C. Among them, high adherence to dietary patterns A and C was associated with a lower risk of myopia. Dietary pattern A was characterized by the consumption of meats, aquatic products, dairy and its products, eggs, and legumes. Dietary pattern C was characterized by the consumption of vegetables, fruits, grains, and potatoes. These associations were independent of sex, age, or other potential confounding factors. In contrast, dietary pattern B, characterized by the consumption of snacks, beverages, and mycorrhizae, was not associated with myopia.

Dietary patterns A and C showed considerable overlap between the characteristics of these foods and the components of other conceptually healthy dietary patterns. Both dietary patterns are characterized by the consumption of foods similar to those found in the Jiangnan diet [27] and Mediterranean diet [28]. The Jiangnan diet includes high consumption of vegetables and fruits in season, fresh-water fish and shrimp, and legumes, as well as moderate consumption of wholegrain rice, plant oils (mainly rapeseed oil), and red meats. Wang et al. [27] reported that the Jiangnan diet is suitable for future national diet recommendations for Chinese people. Additionally, some studies have shown a possible protective effect of the Mediterranean diet against age-related ocular pathologies [29]. For instance, Raimundo et al. [30] believed that fruit consumption in the Mediterranean diet might have protective effects against age-related macular degeneration. This result not only has biological plausibility [31,32] because of the high intake of antioxidants but has also been previously described in the literature in at least one study [33]. 

In several studies, specific meats, fruits, and vegetables have revealed the role of dietary factors in protection against myopia. This protective effect may be attributed to the beneficial components. In 1956, Gardiner indicated that myopes consumed significantly lower amounts of animal protein than non-myopes [34,35]. In addition, he further showed that by increasing the level of animal protein in the diet of myopic children, myopia progressed more slowly than in a control group that did not receive dietary modifications during a one-year experiment [36]. In a trial employing a double-blind, placebo-controlled crossover design, Hitoshi et al. reported that oral intake of a blackcurrant anthocyanoside improved transient myopic shift after prolonged visual tasks and subjective symptoms of visual fatigue in healthy subjects [37]. In an in vitro study, an anthocyanin complex from the bilberry (Vaccinium myrtillus L.) fruit had a relaxing effect on the ciliary muscle, which is important in the treatment of myopia [38]. Recently, a few studies have reported that dietary supplementation with lutein from green leafy vegetables, such as kale and spinach, can increase the content of macular pigments in the retina, thus improving visual function, alleviating vision loss, and repairing damaged retinal tissue to a certain extent [39]. In a cross-sectional study of 4166 people aged 65 and older in Europe, an unexpected finding was that the highest quintile of plasma lutein concentration was associated with a reduced OR of myopia (OR, 0.57; 95% CI, 0.46–0.72) [40]. Hence, diet may be an environmental factor common to both the development of myopia and generalized accelerated growth.

In this study, the overall prevalence of myopia among 10–11-year-old children in China was 23.5%, which increased with age. Compared with boys, girls were more likely to have myopia (26.3% vs. 20.8%), which concurs with the findings of other studies [41]. The prevalence of myopia among urban children is often reported to be higher than that of children from rural groups (31.1% vs. 18.0%), similar to the findings from previous research [42,43]. Meanwhile, our results revealed that those living in northern China were at higher risk of developing myopia than children living in southern China (northern China: 25.7% vs. 21.2%; *p* < 0.001). These differences may be related to dietary differences between the northern and southern regions of China. Southern China has more dark-green leafy vegetables and fruits. Both diets were closer to the Jiangnan diet than the diet of the northern Chinese. Additionally, our observation of a strong association between nutritional status and myopia in children is inconsistent with findings in Korea [44,45] and Ireland [46]. 

The current study covered a wide area with a large sample size and high representativeness, reflecting the distribution characteristics of dietary intake and myopia among 10–11-year-old children in different regions of China. Additionally, our findings showed that dietary patterns A and C were associated with a lower risk of myopia. It is important to provide valuable information for the primary prevention of childhood myopia through dietary modification in Chinese children. However, this study has several limitations. First, we retrospectively collected the habitual intake of a comprehensive range of food items from Chinese children’s exposure data through questionnaires. Thus, we acknowledge that our measure was susceptible to recall bias, similar to many other dietary assessment methods. However, the FFQ covers a longer period of dietary history, varying from months to years, and is the most commonly used method for assessing eating habits [47]. Second, due to the inherent limitations of this cross-sectional study, it was difficult to draw a causal relationship between dietary patterns and myopia. Therefore, we recommend further prospective studies to confirm the association between diet and myopia.

## 5. Conclusions

In this study, we identified three dietary patterns in Chinese children aged 10–11 years at baseline, namely dietary patterns A, B, and C. Among these, dietary patterns A and C had a protective effect against the risk of myopia in Chinese children aged 10–11 years. Both of these dietary patterns are characterized by high consumption of meats, aquatic product, dairy and its products, eggs, legumes, vegetables, fruits, grains, and potatoes. These findings highlight the importance of children’s dietary patterns to myopia and indicate new prospects for myopia prevention through effective dietary interventions.

## Figures and Tables

**Table 1 nutrients-15-01946-t001:** Distribution of demographic characteristics across categories of myopia.

Characteristics	All	Myopia	Non-Myopia	χ2	*p*-Value
Total	7423	1746 (23.5)	5677 (76.5)		
Sex (%)				30.986	<0.001
Boy	3723	774 (20.8)	2949 (79.2)		
Girl	3700	972 (26.3)	2728 (73.7)		
Age (%)				58.717	<0.001
10	3686	727 (19.7)	2959 (80.3)		
11	3737	1019 (27.3)	2718 (72.7)		
Residency (%)				173.819	<0.001
Urban	3128	973 (31.1)	2155 (68.9)		
Rural	4292	771 (18.0)	3521 (82.0)		
Region (%)				20.610	<0.001
Northern China	3770	969 (25.7)	2801 (74.3)		
Southern China	3650	775 (21.2)	2875 (78.8)		
Nutrition status (%)				17.139	<0.001
Stunting and wasting	498	91 (18.3)	407 (81.7)		
Normal	4622	1045 (22.6)	3577 (77.4)		
Overweight and obesity	2181	567 (26.0)	1614 (74.0)		
Medium- and high-intensity physical activity (d/w)				3.173	0.075
0	2392	515 (21.5)	1877 (78.5)		
1	1303	323 (24.8)	980 (75.2)		
2	1828	465 (25.4)	1363 (74.6)		
≥3	1894	443 (23.4)	1451 (76.6)		
Screen time (min/d)				0.010	0.922
0	15	5 (33.3)	10 (66.7)		
>0~59	5190	1217 (23.4)	3973 (76.6)		
≥60	2213	524 (23.7)	1689 (76.3)		
Sleeping time (h/d)				0.192	0.662
≤9	2328	555 (23.8)	1773 (76.2)		
>9	5095	1191 (23.4)	3904 (76.6)		

**Table 2 nutrients-15-01946-t002:** Dietary patterns and factor loadings identified using factor analysis.

Dietary Pattern A		Dietary Pattern B		Dietary Pattern C	
Total Variance: 16.55%		Total Variance: 15.49%		Total Variance: 14.20%	
Food Group	Factor Loading	Food Group	Factor Loading	Food Group	Factor Loading
Meats	0.735	Snacks	0.738	Vegetables	0.807
Aquatic product	0.641	Beverages	0.725	Fruits	0.728
Dairy and its products	0.627	Mycorrhizae	0.421	Grains and potatoes	0.432
Eggs	0.373				
Legumes	0.338				

**Table 3 nutrients-15-01946-t003:** Multivariable adjusted ORs and 95% CI of myopia according to quartiles of dietary pattern adherence.

	Q1	Q2	*p*	Q3	*p*	Q4	*p*	*p* for Trend
Dietary pattern A	*n* = 1857	*n* = 1857		*n* = 1855		*n* = 1854		
Unadjusted	Ref	0.80 (0.69–0.94)	0.006	0.77 (0.66–0.89)	0.001	0.69 (0.59–0.81)	<0.001	<0.001
Model 1	Ref	0.83 (0.71–0.97)	0.022	0.83 (0.71–0.97)	0.021	0.78 (0.66–0.91)	0.002	0.004
Model 2	Ref	0.82 (0.70–0.97)	0.017	0.83 (0.71–0.98)	0.029	0.78 (0.66–0.92)	0.003	0.007
Dietary pattern B	*n* = 1855	*n* = 1857		*n* = 1855		*n* = 1856		
Unadjusted	Ref	1.31 (1.12–1.53)	0.001	1.14 (0.98–1.33)	0.090	0.93 (0.80–1.07)	0.297	0.124
Model 1	Ref	1.30 (1.10–1.52)	0.001	1.15 (0.98–1.34)	0.079	0.99 (0.86–1.16)	0.989	0.613
Model 2	Ref	1.28 (1.09–1.51)	0.002	1.15 (0.98–1.34)	0.089	1.01 (0.87–1.18)	0.916	0.731
Dietary pattern C	*n* = 1855	*n* = 1855		*n* = 1858		*n* = 1855		
Unadjusted	Ref	0.82 (0.70–0.97)	0.016	0.70 (0.60–0.82)	<0.001	0.71 (0.60–0.82)	<0.001	<0.001
Model 1	Ref	0.78 (0.67–0.92)	0.003	0.71 (0.61–0.83)	<0.001	0.69 (0.59–0.81)	<0.001	<0.001
Model 2	Ref	0.76 (0.65–0.90)	0.001	0.70 (0.60–0.82)	<0.001	0.68 (0.58–0.80)	<0.001	<0.001

Model 1 was adjusted for sex, age, residence, and region of origin. Model 2 was additionally adjusted for nutritional status, medium- and high-intensity physical activity, screen time, and sleep duration.

## Data Availability

Data are available upon request owing to privacy restrictions. Data presented in this study are available upon request from the corresponding author. These data are not publicly available because of privacy concerns.

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
