# Peer review of "Dietary Patterns and Associations with Myopia in Chinese Children"

_nutrients, 2023, doi:10.3390/nu15081946_

Round 1

Reviewer 1 Report

1.       In the introduction, the author could write some environmental factors and then explain the dietary.

2.       What is the relationship between Puberty and this study? Is that one of the hypotheses?

3.       Explain the effect of physical activity and sleeping in the result section.

4.       In this study, how the author controls participants' genetic background involves myopia. How the author detects that the food and nutrient intake is closely related to myopia development in selected participants.

5.       Describe the limitation and straightness and suggestions for future study.

6.    What are the home messages of this study?

7.       The discussion section needs to describe and explain the results.

8.       This study is just a report. The authors have to explain how and why. 

Author Response

cover letter

Manuscript ID number: nutrients-2323849

Title of paper: Dietary patterns and associations with myopia in Chinese children

  1. In the introduction, the author could write some environmental factors and then explain the dietary.

Respond: Thank you for your suggestions. We have added the information in the revised manuscript, see words in red in line 52-61.

  1. What is the relationship between Puberty and this study? Is that one of the hypotheses?

Respond: Thank you for your question. Because the population in this study was in Puberty, previous research has found that eye growth may be more vulnerable to environmental factors during puberty. Therefore, this study hypothesizes that this age group is more susceptible to dietary factors.

  1. Explain the effect of physical activity and sleeping in the result section.

Respond: Thank you for your suggestions. You thought that physical activity and sleeping should play a role in myopia. we have done as required, however, no significance was found. This may be due to limitations in the study design.

  1. In this study, how the author controls participants' genetic background involves myopia. How the author detects that the food and nutrient intake is closely related to myopia development in selected participants.

Respond: Thank you for your question. It is regrettable that the genetic background was not included in this study, and we will strengthen this discussion in future studies. In this study, the 11 groups in the FFQ were used to explore dietary patterns using principal component analysis (PCA) with orthogonal (varimax) rotation. The–Kaiser–Meyer Olkin and Bartlett’s tests were used to determine whether the data were suitable for PCA. Three dietary patterns were identified according to factors with eigenvalues greater than 1, combined with the scree plot breakpoints and cumulative variance contributions. Binary logistic regression was used to examine the association between each dietary pattern and myopia. Myopia was used as a dependent variable, and the scores of each dietary pattern in quartiles were used as independent variables.

  1. Describe the limitation and straightness and suggestions for future study.

Respond: Thank you for your suggestions. this study has several limitations. First, we retrospectively collected the habitual intake of a comprehensive range of food items from Chinese children’s exposure data through questionnaires. Thus, we acknowledge that our measure was susceptible to recall bias, similar to many other dietary assessment methods. Second, due to the inherent limitations of this cross-sectional study, it was difficult to draw a causal relationship between dietary patterns and myopia. Therefore, we recommend further prospective studies to confirm the association between diet and myopia. We have wrote the information in the manuscript, see words in line 299-308. 

  1. What are the home messages of this study?

Respond: Thank you for your question. In this study, we identified three dietary patterns in Chinese children aged 10–11 years at baseline: dietary patterns A, B, and C. Among these, dietary patterns A and C had a protective effect against the risk of myopia in Chinese children aged 10–11 years. Both of these dietary patterns are characterized by high consumption of meat, aquatic product, dairy and its products, eggs, legumes, vegetables, fruits, grains, and potatoes. We have wrote the information in the manuscript, see words in line 310-314.

  1. The discussion section needs to describe and explain the results.

Respond: Thank you for your suggestions.

  1. This study is just a report. The authors have to explain how and why.

Respond: Thank you for your suggestions.

Reviewer 2 Report

The work makes a significant contribution to science. Congratulations on the workmanship. The weakest point of the work is the introduction that does not describe the problem sufficiently.

Line 22 and 23 – ‘’ Our findings identified an independent association between dietary patterns and myopia.’’  - In my opinion, this is too much generalization should be specified.

Line 25 - Suggestion for authors. I suggest to add more keywords to their maximum number, this will make the article easier to find.

Line 29 – ‘’ population by 2050[1].’’ - Incorrect form of introduction of quotes. A space at the beginning then a quote. Please correct throughout the text. Should be ‘’ population by 2050 [1].’’.

Line 27 – Introduction - In relation to the entire work, the introduction is too short.

Paragraph 1

·       Please add what is myopia. The definition will help those outside the subject matter understand what the work is about - this increases the viewers .

Subudhi P, Agarwal P. Myopia. [Updated 2022 May 21]. In: StatPearls [Internet]. Treasure Island (FL): StatPearls Publishing; 2023 Jan-. Available from: https://www.ncbi.nlm.nih.gov/books/NBK580529/

·       In which cases we talk about refractive error. What is the range between high and low myopia. 

Flitcroft DI, He M, Jonas JB, Jong M, Naidoo K, Ohno-Matsui K, Rahi J, Resnikoff S, Vitale S, Yannuzzi L. IMI - Defining and Classifying Myopia: A Proposed Set of Standards for Clinical and Epidemiologic Studies. Invest Ophthalmol Vis Sci. 2019 Feb 28;60(3):M20-M30. doi: 10.1167/iovs.18-25957. PMID: 30817826; PMCID: PMC6735818.

·       Effect of changes in axial eyeball length on refractive error

Fan Q, Wang H, Jiang Z. Axial length and its relationship to refractive error in Chinese university students. Cont Lens Anterior Eye. 2022 Apr;45(2):101470. doi: 10.1016/j.clae.2021.101470. Epub 2021 May 22. PMID: 34030907.

·       Risk of myopia-related sequelae.

serious consequences: myopic macular degeneration, retinal detachment, cataract, open angle glaucoma, blindness etc.

Haarman AEG, Enthoven CA, Tideman JWL, Tedja MS, Verhoeven VJM, Klaver CCW. The Complications of Myopia: A Review and Meta-Analysis. Invest Ophthalmol Vis Sci. 2020 Apr 9;61(4):49. doi: 10.1167/iovs.61.4.49. PMID: 32347918; PMCID: PMC7401976.

Williams K, Hammond C. High myopia and its risks. Community Eye Health. 2019;32(105):5-6. PMID: 31409941; PMCID: PMC6688422.

lesser consequences: migraines, musculoskeletal changes, etc.

Harle DE, Evans BJ. The correlation between migraine headache and refractive errors. Optom Vis Sci. 2006 Feb;83(2):82-7. doi: 10.1097/01.opx.0000200680.95968.3e. PMID: 16501409.

ZieliÅ„ski G, Wójcicki M, Rapa M, Matysik-Woźniak A, Baszczowski M, Ginszt M, Litko-Rola M, Szkutnik J, RóżyÅ‚o-Kalinowska I, Rejdak R, Gawda P. Masticatory Muscle Thickness and Activity Correlates to Eyeball Length, Intraocular Pressure, Retinal and Choroidal Thickness in Healthy Women versus Women with Myopia. J Pers Med. 2022 Apr 13;12(4):626. doi: 10.3390/jpm12040626. PMID: 35455742; PMCID: PMC9027064.

Paragraph 2

·       I suggest add possible impact of vitamin D3 levels.

Tideman JW, Polling JR, Voortman T, Jaddoe VW, Uitterlinden AG, Hofman A, Vingerling JR, Franco OH, Klaver CC. Low serum vitamin D is associated with axial length and risk of myopia in young children. Eur J Epidemiol. 2016 May;31(5):491-9. doi: 10.1007/s10654-016-0128-8. Epub 2016 Mar 8. PMID: 26955828; PMCID: PMC4901111.

·       I also suggest to the authors to familiarize themselves with the work :

Liang, Yifan, Shin-ichi Ikeda, Junhan Chen, Yan Zhang, Kazuno Negishi, Kazuo Tsubota, and Toshihide Kurihara. 2023. "Myopia Is Suppressed by Digested Lactoferrin or Holo-Lactoferrin Administration" International Journal of Molecular Sciences 24, no. 6: 5815. https://doi.org/10.3390/ijms24065815

Line 93 - ,, 2.4. Physical activity questionnaire’’ - I suggest moving to the next page.

In my opinion, the description of methods, results, discussions and conclusions is correct.

References - are not written in the style recommended by the journal. They are written in different styles, sometimes missing cited pages (16,17,20,29). Please standardize add the missing pages and change the form to the recommended style.

Author Response

cover letter

Manuscript ID number: nutrients-2323849

Title of paper: Dietary patterns and associations with myopia in Chinese children

The work makes a significant contribution to science. Congratulations on the workmanship. The weakest point of the work is the introduction that does not describe the problem sufficiently.

Respond: Thank you for your comment.

Line 22 and 23 – ‘’ Our findings identified an independent association between dietary patterns and myopia.’’  - In my opinion, this is too much generalization should be specified.

Respond: Thank you for your suggestions. We have changed it in the revised manuscript, see the read words in line 22-23.

Line 25 - Suggestion for authors. I suggest to add more keywords to their maximum number, this will make the article easier to find.

Respond: Thank you for your suggestions. We have added the information in the revised manuscript, see words in red in line 26.

Line 29 – ‘’ population by 2050[1].’’ – Incorrect form of introduction of quotes. A space at the beginning then a quote. Please correct throughout the text. Should be ‘’ population by 2050 [1].’’.

Respond: Thank you for your correction. We have changed it in the revised manuscript, see the read words in line 46-47.

Line 27 – Introduction - In relation to the entire work, the introduction is too short.

Respond: Thank you very much for your suggestion. We have added the information in the revised manuscript.

Paragraph 1

  • Please add what is myopia. The definition will help those outside the subject matter understand what the work is about - this increases the viewers.

Respond: Thank you very much for your suggestion. We have added the information in the revised manuscript, see words in red in line 29-32.

Subudhi P, Agarwal P. Myopia. [Updated 2022 May 21]. In: StatPearls [Internet]. Treasure Island (FL): StatPearls Publishing; 2023 Jan-. Available from: https://www.ncbi.nlm.nih.gov/books/NBK580529/

  • In which cases we talk about refractive error. What is the range between high and low myopia.

Respond: Thank you very much for your suggestion. We have added the information in the revised manuscript, see words in red in line 35-40.

Flitcroft DI, He M, Jonas JB, Jong M, Naidoo K, Ohno-Matsui K, Rahi J, Resnikoff S, Vitale S, Yannuzzi L. IMI - Defining and Classifying Myopia: A Proposed Set of Standards for Clinical and Epidemiologic Studies. Invest Ophthalmol Vis Sci. 2019 Feb 28;60(3):M20-M30. doi: 10.1167/iovs.18-25957. PMID: 30817826; PMCID: PMC6735818.

  • Effect of changes in axial eyeball length on refractive error

Respond: Thank you very much for your suggestion. We have added the information in the revised manuscript, see words in red in line 32-35.

Fan Q, Wang H, Jiang Z. Axial length and its relationship to refractive error in Chinese university students. Cont Lens Anterior Eye. 2022 Apr;45(2):101470. doi: 10.1016/j.clae.2021.101470. Epub 2021 May 22. PMID: 34030907.

  • Risk of myopia-related sequelae.

Respond: Thank you very much for your suggestion. We have added the information in the revised manuscript, see words in red in line 40-45.

serious consequences: myopic macular degeneration, retinal detachment, cataract, open angle glaucoma, blindness etc.

Haarman AEG, Enthoven CA, Tideman JWL, Tedja MS, Verhoeven VJM, Klaver CCW. The Complications of Myopia: A Review and Meta-Analysis. Invest Ophthalmol Vis Sci. 2020 Apr 9;61(4):49. doi: 10.1167/iovs.61.4.49. PMID: 32347918; PMCID: PMC7401976.

Williams K, Hammond C. High myopia and its risks. Community Eye Health. 2019;32(105):5-6. PMID: 31409941; PMCID: PMC6688422.

lesser consequences: migraines, musculoskeletal changes, etc.

Harle DE, Evans BJ. The correlation between migraine headache and refractive errors. Optom Vis Sci. 2006 Feb;83(2):82-7. doi: 10.1097/01.opx.0000200680.95968.3e. PMID: 16501409.

ZieliÅ„ski G, Wójcicki M, Rapa M, Matysik-Woźniak A, Baszczowski M, Ginszt M, Litko-Rola M, Szkutnik J, RóżyÅ‚o-Kalinowska I, Rejdak R, Gawda P. Masticatory Muscle Thickness and Activity Correlates to Eyeball Length, Intraocular Pressure, Retinal and Choroidal Thickness in Healthy Women versus Women with Myopia. J Pers Med. 2022 Apr 13;12(4):626. doi: 10.3390/jpm12040626. PMID: 35455742; PMCID: PMC9027064.

Paragraph 2

  • I suggest add possible impact of vitamin D3 levels.

Respond: Thank you very much for your suggestion. We have added the information in the revised manuscript, see words in red in line 72-77.

Tideman JW, Polling JR, Voortman T, Jaddoe VW, Uitterlinden AG, Hofman A, Vingerling JR, Franco OH, Klaver CC. Low serum vitamin D is associated with axial length and risk of myopia in young children. Eur J Epidemiol. 2016 May;31(5):491-9. doi: 10.1007/s10654-016-0128-8. Epub 2016 Mar 8. PMID: 26955828; PMCID: PMC4901111.

  • I also suggest to the authors to familiarize themselves with the work :

Respond: Thank you very much for your suggestion.

Liang, Yifan, Shin-ichi Ikeda, Junhan Chen, Yan Zhang, Kazuno Negishi, Kazuo Tsubota, and Toshihide Kurihara. 2023. "Myopia Is Suppressed by Digested Lactoferrin or Holo-Lactoferrin Administration" International Journal of Molecular Sciences 24, no. 6: 5815. https://doi.org/10.3390/ijms24065815

  • Line 93 - ,, 2.4. Physical activity questionnaire’’ - I suggest moving to the next page.

 Respond: Thank you very much for your suggestion.

In my opinion, the description of methods, results, discussions and conclusions is correct.

Respond: Thank you for your comment.

  • References - are not written in the style recommended by the journal. They are written in different styles, sometimes missing cited pages (16,17,20,29). Please standardize add the missing pages and change the form to the recommended style.

Respond: Thank you very much for your suggestion. We have changed it in the revised manuscript.

Round 2

Reviewer 1 Report

Thank you.